# Can digital skill protect against job displacement risk caused by artificial intelligence? Empirical evidence from 701 detailed occupations

Ni Chen📷⊙, Zhi Li⊙, Bo Tang📷*⊙

School of Public Policy and Administration, Chongqing University, Chongqing, China

⊙ These authors contributed equally to this work.
* hipytea@gmail.com

**Data Availability Statement:** All relevant data are within the paper and its Supporting Information files.

## Abstract

To identify the role of digital skill in the skill-biased technological changes caused by artificial intelligence, this study estimates the impacts of displacement risk on occupational wage and employment and examines the moderation effects of digital skill through the occupational data from the U.S. Bureau of Labor Statistics through the methods of fixed-effects modeling, heterogeneity analyzing and moderation effect testing. The results highlight three main points that (1) the displacement risk by artificial intelligence has significantly negative effects on occupational wage and employment, (2) the heterogeneous effects across occupational characteristics are significant, and (3) the digital skill exerts a significant moderation effect to protect against displacement risk. The core policy implication is suggested to emphasize digital skill in education and training across occupations to accommodate job requirements in the future.

## 1. Introduction

The rapid development of artificial intelligence (AI) has made enormous contributions to industrial and economic growth over the past decade. However, anxiety about technological unemployment, namely being displaced by AI, has spread globally as technological breakthroughs and revolutions occur [1]. Although the potential risks of AI on society include ethics, security, law and education, AI's impacts on the labor market have been heatedly debated. The relevant debates spell out an increasing concern over human right to work and engage in productive employment and whether AI can displace the roles of the workforce on a larger scale. While various occupations have recognized the potential risks of AI, the relevant debates consider whether the impact of AI on the labor market will be good or bad. On the one hand, the pessimistic voice claims that automation or computerization in the workplace turns into job loss, and many employments face displacement risk [2, 3]. For instance, Acemoglu and Restrepo [4] examined the effects of automation driven by AI on the U.S. labor market and estimated the average reduction of about 0.18% to 0.34% in employments and 0.25% to 0.5%

**Funding:** The authors received no specific funding for this work.

**Competing interests:** The authors have declared that no competing interests exist.

in wages. On the other hand, the optimists state that AI guarantees the quality of economic growth and productivity improvement [5]. Furthermore, more job opportunities will be produced because employment opportunities are continually created in new occupations, such as repairers, conductors, managers, financiers, and new industries [6]. Hence, the impact of AI on the labor market is complex and multi-layered. However, one thing is for sure, the impact of technological advances on labor markets is often inseparable from skill changes in the labor force. Katz and Murphy [7] discuss skill-biased technological change (SBTC), revealing that while labor supply continues to grow, technological progress has significantly affected laborers' employment and wage premiums. Therefore, this study aims to examine the impacts of AI on the labor market to demonstrate the relationships among displacement risk, emerging skills and labor market outcomes to provide empirical evidence for implementing technical and vocational education.

## 2. Literature review

According to the SBTC framework, continuous technological advancement is expected to enlarge industrial demand for highly educated labor and intensify employment inequality [8]. AI advancement causes shocks in occupations, resulting in a reduction in labor demand and wage fluctuations. Therefore, it is suitable that the SBTC framework offers feasible theoretical guidance for examining the impacts of AI on risks, skills and outcomes.

### 2.1. Displacement risk caused by AI

Many approaches, such as task-based, skill-based, and occupation-based methods, help measure the effects of AI displacement. Autor et al. [9] investigated routine-biased technological changes (RBTC) and classified the tasks into two dimensions: cognitive vs. mental and routine vs. non-routine, to estimate displacement risk. Frey and Osborne [3] used a Gaussian process classifier to predict the probability of computerization of 702 occupations and outlined the expected impacts on the labor force. It has been reported that 47% of U.S. employment was automated quickly. Plenty of follow-up research applied the estimated probability of computerization to explore the relationship with labor demands in other sectors. Thereby, the reason why jobs are prone to automation can be uncovered [10]. However, according to Dauth et al. [11], the use of robots altered Germany's employment structure and shrank employment opportunities in the manufacturing industry, while employment increased in the service industry.

### 2.2. Digital skills for occupations

Digital skills resulted from digital literacy, which was considered to appropriately understand and use various digital sources as the modern occupational skills [12]. Deursen and Van Dijk [13] proposed various concepts to account for medium-related content-related skills, comprising operational, formal, information, communication, content creation and strategic skills [14]. In line with such concepts [15], the ability to develop and use information and communication technologies (ICT) is determined by electronically enabled information and the ability to synthesize information into practical and relevant knowledge. In the new educational scenarios, digital skills play a dominant role to transmit quality knowledge in pedagogical processes [16]. Digital skills embody a solution to addressing employment challenges and labor issues caused by technological advancement [17]. It is common to see that digital skills have been indispensable as digital technologies utilized to realize innovations nowadays [18]. Digital technologies are instrumental for daily life and work satisfaction in the digital age, thus the digital skills are more necessary to everyone than ever. According to practical results, the levels of

salaries and wages are strongly correlated with digital proficiency and Internet usage, a consistent effort to increase the digital skills of individuals may be required to achieve a more effective and flexible labor market [19].

## 2.3. Effects of AI on labor market

Past research has explored the impact of new digital technologies on occupations outcomes such as wages and unemployment [20, 21]. Most considered AI the main factor for labor market distribution and polarization [22–24]. The wage difference between high-skilled and low-skilled laborers continues to widen as manifested in the continuous tilt of income distribution to the group of highly educated and skilled laborers. The second view is that AI is an emerging technology that promotes labor productivity [25–28]. Simultaneously, there is an ambiguous view that the impact of AI is not absolute and stable and is constantly changing [29–31].

## 2.4. Summary

The above literature suggests that the technological changes by AI promote economic productivity at the macroeconomic level in the long run, while reducing wages and employment at the individual level in the short run. However, the literature gap shows that less research has investigated occupational effects of AI, even though occupation requirements are critical issues to understanding technological progress and social change, deconstructing human capital in workplaces [32]. Another significant gap in previous research is the lack of attention to digital skills. Skill about AI technologies are in demand to fill skill gaps for the new requirements in most occupations. Therefore, to identify the theoretical relationships of these variables as shown in Fig 1, this study proposes the following hypotheses:

**Hypothesis 1.** Displacement risk will negatively affect occupational wages.

**Hypothesis 2.** Displacement risk will negatively affect occupational employment.

**Hypothesis 3.** Displacement risk and digital skills will have a positive synergistic interaction effect on occupational wage.

**Hypothesis 4.** Displacement risk and digital skills will have a positive synergistic interaction effect on occupational employment.

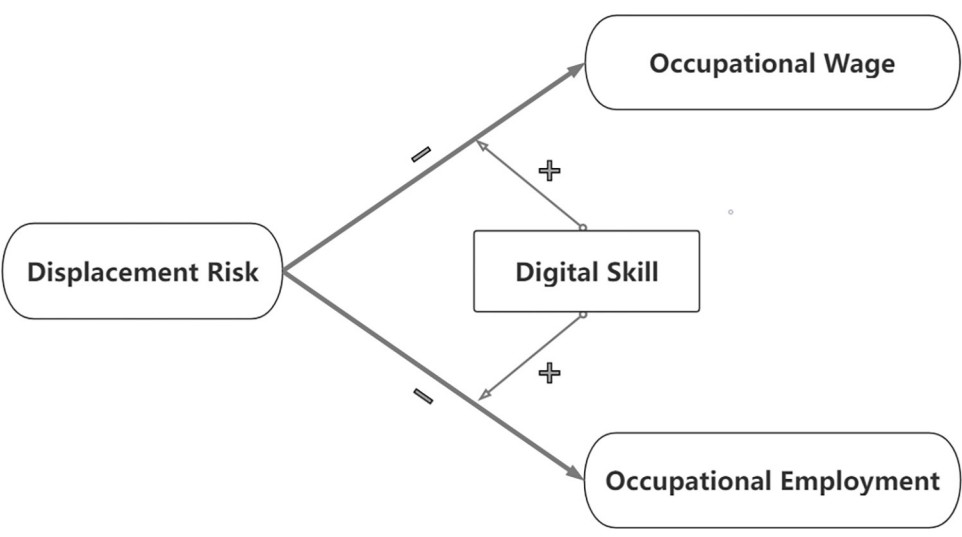

**Fig 1. Theoretical framework and hypotheses.**

## 3. Methods

### 3.1. Empirical model

Considering the above discussion, the following equation is constructed to estimate how displacement risks caused by AI affect occupational wage and employment.

$$Y = \alpha_1 + \beta_1 Risk + \delta_1 Level + \varepsilon$$

A fixed-effect model is employed to describe how displacement risks affect occupational wage and employment after controlling the year and occupational category as fixed effects.

$$Y_{i,t} = \alpha_2 + \beta_2 Risk_{i,t} + \delta_2 Level_{i,t} + Category_i + Year_t + \varepsilon_{i,t}$$

Furthermore, the interaction term (*Risk×Skill*) is adopted into the fixed-effect model to estimate the moderation effect of digital skill.

$$Y_{i,t} = \alpha_3 + \beta_3 Risk_{i,t} + \gamma(Risk \times Skill)_{i,t} + \delta_3 Level_{i,t} + Category_i + Year_t + \varepsilon_{i,t}$$

where *Y* represents the annual wages (*Wage*) and employment (*Employment*) of occupations, both are used as the natural logarithm form in the subsequent estimations.

*Risk* represents displacement risk indicated by the occupation-specific displacement probability by AI.

*Level* represents a comprehensive control variable, including the main occupation characteristics clarified into five levels based on the requirements of education, experience and training.

*Skill* represents the digital skill requirement of occupations needed to successfully perform a job, which is the moderation variable indicated by the types of software used.

*Category* is the major classification of the Standard Occupational Classification (SOC), which classifies occupations at four levels of aggregation: major, minor, broad, detailed; *Year* is period. Parameters *i* and *t* denote the occupation belonging to the specific category and year. Consolidated.

### 3.2. Data collection

**3.2.1. Stage 1.**    Collecting the raw data: This study collected data of indicators mentioned above from several publicly available datasets. First, the labor market outcome data of *Wage* and *Employment* were obtained from the Occupational Employment Statistics (OES) database on the website of U.S. Bureau of Labor Statistics. Second, the occupational characteristic data of *Level* and *Category* were collected from the Occupational Information Network (O*NET) database on the website of O*NET On-line. Third, the displacement risk data of *Risk* was from the estimated results by Frey and Osborne [3], who implemented the Gaussian process classification methodology to estimate the automation probability for 702 detailed occupations. All the probability results for all detailed occupations are presented in the S1 and S2 Tables and S1 Dataset of the article.

**3.2.2. Stage 2.**    Consolidating the selected datasets: This study consolidated the selected datasets followed the principle of data availability, those samples with missing critical data were excluded. The selected datasets were matched according to the O*NET-SOC taxonomy to form the final dataset which is composed of 4,907 observations referring to 701 detailed occupations from 2013 to 2019.

**3.2.3. Stage 3.**    Describing the dataset statistics: This study conducted the descriptive statistics of the final dataset. Table 1 summarizes the variable statistics. Overall, all the data used in this study has been uploaded for sharing as the S1 Dataset of this article.

**Table 1. Descriptive statistics of the variables.**

| Sign | N | Mean | SD | Min. | Max. |
|---|---|---|---|---|---|
| Wage ($) | 4907 | 55,233 | 26,603 | 18,870 | 242,740 |
| Employment (person) | 4907 | 159,603 | 404,467 | 290 | 4612,510 |
| Risk (%) | 4907 | 53.609 | 36.781 | 0.280 | 99 |
| Level (No.) | 4907 | 2.944 | 1.097 | 1 | 5 |
| Skill (items) | 4907 | 29.422 | 42.216 | 0 | 404 |

Note: *Level* is measured as the ordinal categorical variables (1 for "occupations that need little or no preparation", 2 for "occupations that require some preparation", 3 for "occupations that need medium preparation", 4 for" occupations that need considerable preparation" and 5 for "occupations that need extensive preparation").

## 3.3. Statistical analysis

Various statistical analysis methods were used in this study, depending on the purpose of the analysis. First, the fixed effects model (FE) was used for the basic regression to directly estimate the effects of the displacement risk on occupational wage and employment. Second, the heterogeneity among occupations was examined by applying the grouped regression method. Finally, the moderation effects of digital skill were analyzed by employing interactions into regressions. STATA 16.0 was employed for all the statistical analyses.

# 4. Results

## 4.1. Basic regression

The regression results shown in Table 2 demonstrate that displacement risk negatively impacted occupational wage (−0.0022, $p < 0.01$) and occupational employment (−0.0055, $p < 0.01$) after controlling for occupational level, category and year as fixed effects. The estimated results suggest that a 1% increase in displacement risk was associated with a 0.22% decrease in occupational wage or a 0.55% decrease in occupational employment. These results also mean that the higher the displacement risk of occupation, the lower the occupational wage and employment. Therefore, Hypotheses 1 and 2 are confirmed.

**Table 2. Basic regression results.**

| | log of Wage | log of Employment |
|---|---|---|
| Risk | −0.0022*** | −0.0055*** |
| | (0.0002) | (0.0009) |
| Level | 0.2082*** | −0.2099*** |
| | (0.0071) | (0.0362) |
| Constant | 10.5716*** | 12.3569*** |
| | (0.0384) | (0.1891) |
| Category | YES | YES |
| Year | YES | YES |
| Adj. $R^2$ | 0.6151 | 0.5333 |
| F | 213.4787 | 536.8086 |
| N | 4907 | 4907 |

Note

*** $p < 0.01$

** $p < 0.05$, and

* $p < 0.1$; the standard error is shown in parentheses under the coefficient.

**Table 3. Estimation results in different occupational levels.**

|  | log of Wage | log of Employment |
|---|---|---|
| Occupation Level 1 (N = 217) | 0.0020 | −0.0028 |
|  | (0.0018) | (0.0107) |
| Occupation Level 2 (N = 1904) | −0.0015*** | 0.0015 |
|  | (0.0004) | (0.0023) |
| Occupation Level 3 (N = 1260) | −0.0020*** | −0.0058*** |
|  | (0.0002) | (0.0018) |
| Occupation Level 4 (N = 987) | −0.0019*** | −0.0033 |
|  | (0.0003) | (0.0021) |
| Occupation Level 5 (N = 539) | −0.0039*** | −0.0069 |
|  | (0.0009) | (0.0045) |

Note

*** $p < 0.01$

** $p < 0.05$, and

* $p < 0.1$; the standard error is shown in parentheses under the coefficient.

## 4.2. Heterogeneity analysis

As occupations have hierarchical and differentiated features, the heterogeneity analysis is necessary to determine the effects of displacement risk in the different groups of occupational levels and categories.

First, according to the occupational levels classified in O*NET, the high-level occupations require more professional knowledge, experience, and skills than low-level occupations. As shown in Table 3, the influence of displacement risk on occupational wage was significantly negative in Level 2 (−0.0015, $p < 0.01$), Level 3 (−0.0020, $p < 0.01$), Level 4 (−0.0019, $p < 0.01$) and Level 5 (−0.0039, $p < 0.01$), while the influence of displacement risk on occupational employment was only significant in Level 3 (−0.0058, $p < 0.01$). The results demonstrate the heterogeneous effects of displacement risk are mainly on occupational wage, suggesting that AI has a greater impact on occupational wages in higher occupational level.

Second, according to the major occupational classification in O*NET, there are 22 categories of occupations classified in this study. The estimation results of occupational categories with wage and employment both-affected are shown in Table 4. The influence of displacement risk on occupational wage and employment were significant in the occupations categories including Arts, Design, Entertainment, Sports, and Media (−0.0027, $p < 0.01$; −0.0066, $p < 0.1$), Business and Financial Operations (−0.0020, $p < 0.01$; −0.0088, $p < 0.05$), Computer and Mathematical (−0.0049, $p < 0.01$; −0.0213, $p < 0.05$), Construction and Extraction (−0.0047, $p < 0.01$; −0.0156, $p < 0.01$), Educational Instruction and Library (−0.0033, $p < 0.01$; −0.0155, $p < 0.05$), Food Preparation and Serving Related (−0.0070, $p < 0.01$; 0.0372, $p < 0.01$), Healthcare Practitioners and Technical (−0.0070, $p < 0.01$; 0.0117, $p < 0.01$), Installation, Maintenance, and Repair (−0.0037, $p < 0.01$; −0.0149, $p < 0.01$), Life, Physical, and Social Science (−0.0041, $p < 0.01$; 0.0048, $p < 0.1$). The results have revealed heterogeneous effects of displacement risk on occupational wage and employment in the different occupational categories. In addition, the full list of estimation results could be seen in S1 Table.

## 4.3. Moderation effect

Regarding whether digital skills can moderate the negative impacts of displacement risk on occupational wage and employment, interaction (*Risk×Skill*) was introduced into the

**Table 4. Estimation results in different major occupational categories.**

| Major Occupational Categories | log of Wage | log of Employment |
|---|---|---|
| Arts, Design, Entertainment, Sports, and Media (N = 231) | −0.0027*** | −0.0066* |
| | (0.0006) | (0.0035) |
| Business and Financial Operations (N = 210) | −0.0020*** | −0.0088** |
| | (0.0005) | (0.0039) |
| Computer and Mathematical (N = 119) | −0.0049*** | −0.0213** |
| | (0.0005) | (0.0097) |
| Construction and Extraction (N = 392) | −0.0047*** | −0.0156*** |
| | (0.0006) | (0.0059) |
| Educational Instruction and Library (N = 154) | −0.0033*** | −0.0155** |
| | (0.0007) | (0.0060) |
| Food Preparation and Serving Related (N = 112) | −0.0070*** | 0.0372*** |
| | (0.0009) | (0.0114) |
| Healthcare Practitioners and Technical (N = 308) | −0.0070*** | 0.0117*** |
| | (0.0008) | (0.0043) |
| Installation, Maintenance, and Repair (N = 350) | −0.0037*** | −0.0149*** |
| | (0.0005) | (0.0046) |
| Life, Physical, and Social Science (N = 294) | −0.0041*** | 0.0048* |
| | (0.0006) | (0.0028) |

Note

*** $p < 0.01$

** $p < 0.05$, and

* $p < 0.1$; the standard error is shown in parentheses under the coefficient.

regression model for re-examination. Moreover, considering that digital skills are highly correlated with occupations requiring a science, technology, engineering, and mathematics (STEM) background or occupations belonging to computers and mathematics, it is necessary to examine the differences and details by sub-grouping. Table 5 reports all the results.

In general, the estimation results of the full sample in Columns (1) and (6) show that the interaction of displacement risk and digital skill significantly impacted occupational wage (0.0002, $p < 0.01$) and employment (0.0068, $p < 0.01$), suggesting that digital skill can play a counteraction role against displacement risk. Thus, digital skills can positively moderate the negative impacts of displacement risk on occupational wage and employment. Thus, Hypotheses 3 and 4 are confirmed.

Specifically, the results of the non-STEM groups in Column (3) and Column (8) show that the interactions were significantly positive on occupational wage (0.0002, $p < 0.01$) and occupational employment (0.0072, $p < 0.01$). Moreover, the interactions of non-IT groups in Columns (5) and (10) were also significantly positive on occupational wage (0.0003, $p < 0.01$) and employment (0.0083, $p < 0.01$). The moderation effect of digital skills was clear in non-STEM and non-IT occupations. The implication may be that digital tools and IT technology can significantly improve work efficiency in these occupations. Therefore, digital skills have competitive advantages, leading to an increase in occupational wages and employment.

## 4.4. Robustness tests

Most related studies usually have used the displacement probability estimated by Frey and Osborne. To make the results more reliable, we decide to test the robustness of the results

**Table 5. Estimation results of moderation effects.**

| | log of Wage | | | | | log of Employment | | | | |
|---|---|---|---|---|---|---|---|---|---|---|
| | **Full** | **STEM** | **Non-STEM** | **IT** | **Non-IT** | **Full** | **STEM** | **Non-STEM** | **IT** | **Non-IT** |
| | **(1)** | **(2)** | **(3)** | **(4)** | **(5)** | **(6)** | **(7)** | **(8)** | **(9)** | **(10)** |
| *Risk* | −0.0027*** | 0.0005 | −0.0026*** | −0.0056*** | −0.0027*** | −0.0193*** | −0.0326 | −0.0191*** | −0.0249* | −0.0205*** |
| | (0.0002) | (0.0022) | (0.0002) | (0.0014) | (0.0002) | (0.0014) | (0.0201) | (0.0015) | (0.0139) | (0.0013) |
| *Risk×Skill* | 0.0002*** | −0.0008 | 0.0002*** | 0.0013 | 0.0003*** | 0.0068*** | 0.0082 | 0.0072*** | −0.0139 | 0.0083*** |
| | (0.0001) | (0.0008) | (0.0001) | (0.0012) | (0.0001) | (0.0005) | (0.0067) | (0.0005) | (0.0096) | (0.0005) |
| *Level* | 0.2043*** | 0.1985*** | 0.2046*** | 0.0614** | 0.2049*** | −0.3213*** | −0.5570*** | −0.3019*** | −1.9324*** | −0.2508*** |
| | (0.0074) | (0.0220) | (0.0079) | (0.0237) | (0.0065) | (0.0362) | (0.0970) | (0.0391) | (0.2818) | (0.0307) |
| *Constant* | 10.5855*** | 10.8171*** | 10.5602*** | 11.1836*** | 10.2554*** | 12.7516*** | 14.3074*** | 12.6084*** | 20.3140*** | 11.8623*** |
| | (0.0391) | (0.1172) | (0.0413) | (0.1048) | (0.0282) | (0.1894) | (0.5048) | (0.2004) | (1.2550) | (0.1387) |
| *Category* | YES | YES | YES | YES | YES | YES | YES | YES | NO | NO |
| *Year* | YES | YES | YES | YES | YES | YES | YES | YES | YES | YES |
| *Adj. R²* | 0.6155 | 0.5324 | 0.5956 | 0.6662 | 0.4794 | 0.5517 | 0.6382 | 0.5494 | 0.7077 | 0.4877 |
| *F* | 207.5462 | 47.3272 | 159.5640 | 18.7182 | 387.6681 | 500.7327 | 181.6218 | 460.7728 | 26.0449 | 1997.2415 |
| *N* | 4907 | 469 | 4438 | 119 | 4788 | 4907 | 469 | 4438 | 119 | 4788 |

Note

*** $p < 0.01$

** $p < 0.05$, and

* $p < 0.1$; the standard error is shown in parentheses under the coefficient.

using different measurement of displacement risk. According to Manyika et al. [33] found some sectors have more automatable activities than others after comparing the automation potential of 19 selected sectors such as manufacturing, agriculture, real estate, educational services and so on. Hence, we take their automation potential as the alternative measurement of displacement risk. Afterward, we manually selected 3 to 8 representative detailed occupations for each of 19 sectors through matching the most relative occupation title and job content. Finally, 66 representative occupations were selected for robustness testing. The alternative displacement risk was presented in Table 6, and the more details could be seen in S2 Table.

The estimation results of robustness tests as shown in Table 7 demonstrate that displacement risk negatively impacted occupational wage (−0.0288, $p < 0.01$) represented in Column (1), but not significantly on occupational employment represented in Column (3). The estimation results of the moderation effects in Columns (2) and (4) show that the new interaction of

**Table 6. The alternative measurement of displacement risk by sectors.**

| Sectors | Risk_d | Sectors | Risk_d |
|---|---|---|---|
| Accommodation and food services | 0.73 | Finance and insurance | 0.43 |
| Manufacturing | 0.6 | Arts, entertainment, and recreation | 0.41 |
| Transportation and warehousing | 0.6 | Real estate | 0.4 |
| Agriculture | 0.57 | Administrative | 0.39 |
| Retail trade | 0.53 | Health care and social assistances | 0.36 |
| Mining | 0.51 | Information | 0.36 |
| Other services | 0.49 | Professionals | 0.35 |
| Construction | 0.47 | Management | 0.35 |
| Utilities | 0.44 | Educational services | 0.27 |
| Wholesale trade | 0.44 | | |

**Table 7. Estimation results of robustness tests.**

|  | Log of Wage | | log of Employment | |
| --- | --- | --- | --- | --- |
|  | (1) | (2) | (3) | (4) |
| *Risk* | −0.0288*** | −0.0297*** | −0.0077 | −0.0089 |
|  | (0.0014) | (0.0014) | (0.0056) | (0.0059) |
| *Risk×Skill* |  | 0.0013*** |  | 0.0015 |
|  |  | (0.0005) |  | (0.0019) |
| *Controls* | YES | YES | YES | YES |
| *Adj. $R^2$* | 0.4640 | 0.4713 | 0.6347 | 0.6344 |
| *F* | 73.8832 | 71.2225 | 728.8624 | 670.9145 |
| *N* | 462 | 462 | 462 | 462 |

Note

*** $p < 0.01$

** $p < 0.05$, and

* $p < 0.1$; the standard error is shown in parentheses under the coefficient.

displacement risk and digital skill significantly impacted occupational wage (0.0013, $p < 0.01$), but not significantly on occupational employment. Thus, after replacing the measurement of displacement risk, only the significant impact on occupational wage remains unchanged, the impacts on occupational employment are not robust.

The above results are possible to be explained in two ways. On the one hand, it is demonstrated that the phenomenon of wage losses for displaced workers caused by technological progress and wage premiums arose from emerging skills has already happened among many sectors [34]. On the other hand, it may be the reason U-shaped employment distribution is known as job polarization [23], which makes the effect of AI's displacement risk on occupational employment failing to show significance in the linear regression.

## 5. Discussions

This study presents empirical evidence from the United States over 2013–2019 to examine the occupational effects of AI. (1) The first finding shows that both occupational wages and employment have been negatively impacted by the displacement risk of AI, which reveals that occupations with higher displacement risk have a reduction in wages and employment. There are similar conclusions in the previous literature. Autor et al. [35] found that labor income in the United States declined in the 1980s and reported that the emergence of automation reduces labor income in industrialized countries. (2) The second finding illustrates that heterogeneous effects have been confirmed across occupational levels and categories. It was shown that the impacts of AI on occupational wages depended on workforce type [36]. Autor and Salomons [37] proposed that the employment polarization rendered by AI demands more low-skilled jobs, which means that AI negatively impacts high-skilled and intensive skilled laborers' wages. (3) The last finding identified the positive moderation effects of digital skills. The role of ICT or digital skills in the present digital economy should not be underestimated [38]. Digital skills could drive organizations' competitiveness and innovation capacity, which are more required to the current economic and social developments in the 21st-century [39]. AI has changed the essence of work, making digital skills a fundamental requirement of the modern labor force [40]. Thus, most occupations try to attract employees with digital skills to adapt to the increasingly digital environment. For example, the professionals in the healthcare sector with higher digital skills can provide better quality of patient treatment and more cost-

effectiveness of work due to their more efficiency on information and technology [41]. Additionally, digital skills appear to improve the individual's labor market opportunities, while 14.5% of workers with higher digital skills are changing jobs more often than 10.3% of those with fewer digital competences, according to the data in the occupation category of "hospitality, retail and other services managers" which is one of occupations at high risk of being automated [42].

Given the above findings, several policies are recommended. First, basic income supports could be established for perceivable unemployment by AI. The application of AI is bound to affect labor markets, and thus social security must concern the weak low-skilled labor as well as building the strong bottom line for whole labor force. Basic income support could provide a comprehensive social safety net against the risk of AI shock. Second, active labor market policies (ALMPs) by Calmfors [43] are another set of policy tools for countervailing the undesirable effect of AI displacement and reducing inequality and negative externalities. Bonoli [44] suggests that ALMPs increase the employment probability for low-skilled workers with basic education or vocational training and improve labor supply quality. In particular, the emergence of online platform data for ALMPs in recent years, which plays a critical role in closing the digital skill gaps such as skill training, job searching and so on [45]. According to the investigation, it is revealed that 76% of OECD and EU countries developing online training and 70% introducing new online courses for designing ALMPs for the recovery [46]. Meanwhile, online freelancing has increasingly become one important strategy of ALMPs for employment promotion, one-third of workers stem from India and 42% of all work stems from software development and tech work according to Online Labour Index 2020 [47]. Finally, filling the digital skills gap in the digital transformation plays an integral part in adapting occupational displacement trends. It is necessary to strengthen digital skills training at multiple levels of basic, professional and vocational education and to accelerate skill transformation to the new era of AI.

The study is subject to some limitations. Unobserved effects perhaps exist in this study due to the limited data of specific individuals. This study only analyzes the effect of displacement risk on wages and employment from the occupational perspective and ignores worker heterogeneity. Therefore, the results can overestimate the occupational impacts of displacement risk. Furthermore, the impacts of AI on occupations are multifaceted. Thus, another study limitation is in only discussing occupational wage and employment, while other important consequences could be deeply concerning in the future.

## 6. Conclusions

In summary, the increasingly widespread AI applications in the future will significantly impact occupations and working patterns, and digital skills will be the bridge for human labor and AI to work together. More importantly, the displacement by AI is not a purely technical subject. It is necessary to strengthen social and ethical discussions surrounding AI.

## Supporting information

**S1 Table. The full list of estimation results.**
(DOCX)

**S2 Table. The full list of alternative measurements.**
(DOCX)

**S1 Dataset. The whole dataset for this study.**
(XLS)

## Author Contributions

**Conceptualization:** Ni Chen, Zhi Li.

**Data curation:** Bo Tang.

**Formal analysis:** Bo Tang.

**Funding acquisition:** Zhi Li.

**Investigation:** Ni Chen.

**Methodology:** Bo Tang.

**Project administration:** Bo Tang.

**Resources:** Bo Tang.

**Software:** Bo Tang.

**Supervision:** Zhi Li.

**Validation:** Ni Chen, Zhi Li, Bo Tang.

**Visualization:** Bo Tang.

**Writing – original draft:** Ni Chen.

**Writing – review & editing:** Zhi Li.

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
