## [Decision Letter · Decision Letter 0]

12 Aug 2022

PONE-D-22-19557Can Digital Skill Protect against Job Displacement Risk Caused by Artificial Intelligence? Empirical Evidence from 701 Detailed OccupationsPLOS ONE

Dear Dr. Bo Tang,

Thank you for submitting your manuscript to PLOS ONE. After careful consideration, we feel that it has merit but does not fully meet PLOS ONE’s publication criteria as it currently stands. Therefore, we invite you to submit a revised version of the manuscript that addresses the points raised during the review process.

We look forward to receiving your revised manuscript.

Kind regards,

Carlos Alberto Zúniga-González, Ph.D

Academic Editor

PLOS ONE

Journal Requirements:

2. In order to meet journal requirements for reporting and reproducibility, at this time we request that you please update the Methods section to report the original source of the data and the methods used to collect it in sufficient detail for another researcher to access the same data in the same manner. Please ensure that you include a statement specifying whether the collection and analysis method complied with the terms and conditions of the data source.

Additional Editor Comments:

Dear authors, the manuscript is interesting with Digital Skill Protect against Job Displacement Risk Caused by Artificial Intelligence?, I would like you clarify the question of reviewers. I suggest you some reference that may to add in introduction, review literature.

[1] Zúniga-Gonzalez, C.A. (2021). The role of the mediator and the student in the new educational scenarios: COVID-19. Electronic Journal Quality in Higher Education, 12(2), 279 - 294.

https://doi.org/10.22458/caes.v12i2.3730

[2] Bermúdez-León, D. S., & Zúniga-Gonzalez, C. A. (2016). Information and communication technologies (ICT) in response to educational needs in rural areas in Nicaragua. Rev. Iberoam. Bioecon. Climate Change, 2(4), 563–574. https://doi.org/10.5377/ribcc.v2i4.5931

[3] Blanco-Orozco, N., Arce-Díaz, E., & Zúñiga-Gonzáles, C. (2015). Integral assessment (financial, economic, social, environmental and productivity) of using bagasse and fossil fuels in power generation in Nicaragua. Revista Tecnología en Marcha, 28(4), 94-107. DOI 10.18845/tm.v28i4.2447 https://publons.com/publon/32281799/

[4] Zuniga González, C. A. (2020). Total factor productivity growth in agriculture: Malmquist index analysis of 14 countries, 1979-2008. REICE: Revista Electrónica De Investigación En Ciencias Económicas, 8(16), 68–97. https://doi.org/10.5377/reice.v8i16.10661

[5] Zuniga-Gonzalez, Carlos Alberto (2021), “Total factor productivity in the INTA Chinandega rice variety”, Mendeley Data, V2, doi: 10.17632/76m7p7mvsg.2 https://data.mendeley.com/datasets/76m7p7mvsg/2

[6] González, C. A. Z. (2011). Technical efficiency of organic fertilizer in small farms of Nicaragua: 1998-2005. African Journal of Business Management, 5(3), 967-973. https://publons.com/publon/11272633/

[7] Dios-Palomares, R., Alcaide, D., Diz, J., Jurado, M., Prieto, A., Morantes, M., & Zúñiga, C. A. (2015). Analysis of the efficiency of farming systems in Latin America and the Caribbean considering environmental issues. Revista Cientifica, Facultad de Ciencias Veterinarias, Universidad del Zulia, 25(1), 43-50. https://publons.com/publon/3106827/

Reviewers' comments:

Reviewer's Responses to Questions

**Comments to the Author**

1. Is the manuscript technically sound, and do the data support the conclusions?

Reviewer #1: Yes

Reviewer #2: Yes

2. Has the statistical analysis been performed appropriately and rigorously? 

Reviewer #1: I Don't Know

Reviewer #2: Yes

3. Have the authors made all data underlying the findings in their manuscript fully available?

Reviewer #1: Yes

Reviewer #2: Yes

4. Is the manuscript presented in an intelligible fashion and written in standard English?

Reviewer #1: Yes

Reviewer #2: Yes

5. Review Comments to the Author

Reviewer #1: Q1: The authors clearly lay out the main hypotheses of the article, and detail the methodology used to test each of these.

Q2: “I don’t know” was selected because labor statistics fall outside the reviewer’s area of expertise. However, it is noted that the authors' statistical methods and underlying literature are explained in detail, and that for added rigor, authors conducted additional analysis with alternative displacement risk measures to confirm robustness of results. Some of the statistical methods and rationale (e.g. robustness analyses) are described in the results section; it would be good to rearrange the sections so that all analytical methods are presented together. It may also be helpful to shorten some of the tables (e.g. table 4) to focus on the most-affected categories, and include the full list in supplements.

Q3: Key variables from data sources are included in tables, and publicly-available data sources have been listed.

Q4:

The introduction and literature review sections are clear and compelling, even to an audience with limited expertise in the fields of AI and labor. However, the discussion and conclusion sections were less grounded in practical implications of the research. Given the relevance of this paper, it would be good to strengthen these sections by expounding on some of the recommendations:

- For example, a key recommendation is the need to develop more digital skills, and it is stated that the potential impact of these “cannot be under-estimated”. From the literature reviewed, could you provide a couple of examples to illustrate the effect of digital skills on occupational impacts of displacement risk from AI, especially for the most affected sectors?

- Active labor market policies are another example of a recommendation that would be enhanced by data-based examples, even if drawn from a limited number/span of studies.

- Lack of individual level data prevents understanding of other social factors that may affect findings and recommendations, as highlighted in the discussion and conclusion sections. These factors might also contribute to heterogeneity. If such data is available for any of the sectors studied, a deeper dive into this aspect could help to illustrate the interplay between social factors and observed impacts of displacement risk on occupational wages and employment.

- It would be good to discuss the findings of the robustness analysis.

- Minor copy-edits in a few places are needed.

This was an interesting and educational paper; the reviewer is keen to reflect more on its applications to their field of work (health) and looks forward to reading the published manuscript in future.

Reviewer #2: The author of the article presented showed his methodology and database and obtained results that compared his hypotheses. Specifically. The author found that the regression results demonstrate that displacement risk negatively impacted occupational wage and occupational employment;suggesting that AI has a greater impact on occupational wages in higher occupational level.Therefore, I believe that the information presented has an appropriate methodological support for its publication.

6. PLOS authors have the option to publish the peer review history of their article (what does this mean?). If published, this will include your full peer review and any attached files.

Reviewer #1: **Yes: **Paula Ihozo Akugizibwe

Reviewer #2: **Yes: **Napoleon Vicente Blanco Orozco

---

## [Author Response · Author response to Decision Letter 0]

24 Aug 2022

1.Response to Journal Requirements and Editor Comments:

Q1: Please ensure that your manuscript meets PLOS ONE's style requirements, including those for file naming.

Response 1: Thanks for the reminder. We have carefully read PLOS ONE's style requirements once again, and then we have checked the manuscript throughout to correct the inappropriate points. All changes have been marked so that the editor and reviewers could see and understand them straightly.

Q2: In order to meet journal requirements for reporting and reproducibility, at this time we request that you please update the Methods section to report the original source of the data and the methods used to collect it in sufficient detail for another researcher to access the same data in the same manner. Please ensure that you include a statement specifying whether the collection and analysis method complied with the terms and conditions of the data source.

Response 2: Thanks for the recommendations. We have tried to make a rearrangement in this section of “3.Methods”. And we have reported the original source of the data used in this article, namely the public website of the databases. In addition to make our data collection more transparent and reproducibility, we have reported our stages of the data collection, and have uploaded all the data used in this study for sharing as in the file “S1 Dataset”. [Lines 162-183]

Q3: Please review your reference list to ensure that it is complete and correct. If you have cited papers that have been retracted, please include the rationale for doing so in the manuscript text, or remove these references and replace them with relevant current references. Any changes to the reference list should be mentioned in the rebuttal letter that accompanies your revised manuscript. If you need to cite a retracted article, indicate the article’s retracted status in the References list and also include a citation and full reference for the retraction notice.

Response 3: Thanks for the reminder. We have double-checked and corrected the in-text citations and end-text references list as the reviewer requested. In this revised manuscript, we have cited several new references as following:

16.Zúniga-Gonzalez, C.A. (2021). The role of the mediator and the student in the new educational scenarios: COVID-19. E J Qual High Educ. 2021;12(2), 279– 294. doi: 10.22458/caes.v12i2.3730.

34.Burda MC, Mertens A. Estimating wage losses of displaced workers in Germany. Labour Econ. 2001;8(1):15-41. doi: 10.1016/s0927-5371(00)00022-1.

41.Jimenez G, et al. Digital health competencies for primary healthcare professionals: A scoping review. Int J Med Inform. 2020;143(104260):104260. doi: 10.1016/j.ijmedinf.2020.104260.

42.Pichler D, Stehrer R. Breaking Through the Digital Ceiling: ICT Skills and Labour Market Opportunities. The Vienna Institute for International Economic Studies. 2021. Available from: https://www.econstor.eu/handle/10419/240636

45.Stephany F. Closing the Digital Skill Gap : The Potential of Online Platform Data For Active Labour Markets Policies. Zenodo. 2022. Available from: https://zenodo.org/record/6684547/files/Stephany-%20Closing%20the%20Digital%20Skill%20Gap.pdf?download=1

46.Organisation for Economic Co-operation and Development. Designing Active Labour Market Policies for the Recovery. OECD Publishing; 2021.

47.Stephany F, Kässi O, Rani U, Lehdonvirta V. Online Labour Index 2020: New ways to measure the world’s remote freelancing market. Big Data Soc. 2021;8(2):205395172110432. doi: 10.1177/20539517211043240.

We have not cited any papers that have been retracted, and we have tried our best to correct any errors we could find. However, if there are still inaccuracies, please point them out and allow us to correct. 

Q4: Additional Editor Comments that suggesting some reference to add in introduction, review literature.

Response 4: Thanks for the recommendation. We have seriously read the recommended literature and cited some of them that are most relevant to our research topic in this article. 

2.Response to Reviewer 1 Comments

Q1: The authors clearly lay out the main hypotheses of the article, and detail the methodology used to test each of these.

Response 1: Thanks for the reviewer’s positive comment.

Q2: “I don’t know” was selected because labor statistics fall outside the reviewer’s area of expertise. However, it is noted that the authors' statistical methods and underlying literature are explained in detail, and that for added rigor, authors conducted additional analysis with alternative displacement risk measures to confirm robustness of results. Some of the statistical methods and rationale (e.g. robustness analyses) are described in the results section; it would be good to rearrange the sections so that all analytical methods are presented together. It may also be helpful to shorten some of the tables (e.g. table 4) to focus on the most-affected categories, and include the full list in supplements.

Response 2: Thanks for the recommendations. We have added one separate section named “3.3.Statistical Analysis” in “3.Methods” to describe all the statistical analysis methods used in this study, including fixed effects model, grouped regression method and moderation effect testing. And we have reported that the statistical software STATA 16.0 was employed for all the statistical analyses. [Lines 188-195] 

Then, we have followed the reviewer's suggestion to shorten the Table 4 and Table 6. After careful consideration, we decided to retain the occupational categories with significant effects both on occupational wage and employment [Lines 241]. In addition, we presented only the alternative measurement of displacement risk by sectors [Lines 282]. The detailed full list of tables was placed in “supporting information” of this revised manuscript.

Q3: Key variables from data sources are included in tables, and publicly-available data sources have been listed.

Response 3: Thanks for the reviewer’s positive comment.

Q4: The introduction and literature review sections are clear and compelling, even to an audience with limited expertise in the fields of AI and labor. However, the discussion and conclusion sections were less grounded in practical implications of the research. Given the relevance of this paper, it would be good to strengthen these sections by expounding on some of the recommendations:

- For example, a key recommendation is the need to develop more digital skills, and it is stated that the potential impact of these “cannot be under-estimated”. From the literature reviewed, could you provide a couple of examples to illustrate the effect of digital skills on occupational impacts of displacement risk from AI, especially for the most affected sectors?

Response 4.1: Thanks for the recommendations. We have listed the examples of the “healthcare sector” and the “hospitality, retail and other services managers” occupation category, to illustrate the benefits of improving digital skills to protect workers in the two most-affected occupational domains by AI [Lines 322-330]. 

- Active labor market policies are another example of a recommendation that would be enhanced by data-based examples, even if drawn from a limited number/span of studies.

Response 4.2: Thanks for the recommendations. We have added some specific and data-based examples as the reviewer’s suggestion, one of which is the online platform data for ALMPs in recent years while playing a critical role in closing the digital skill gaps such as skill training, job searching and so on [Lines 340-348].

- Lack of individual level data prevents understanding of other social factors that may affect findings and recommendations, as highlighted in the discussion and conclusion sections. These factors might also contribute to heterogeneity. If such data is available for any of the sectors studied, a deeper dive into this aspect could help to illustrate the interplay between social factors and observed impacts of displacement risk on occupational wages and employment.

Response 4.3: Thanks for the comments and suggestions. It is a great idea that the analysis of the displacement risk on occupational wages and employment via individual level data. In the process of this study, we had thought about detailed analyzing the impact of AI on the individual among occupations as you mentioned. However, the issues may require a separate article to deal with, we had to give up going further on this direction. We believe that the individual level data could help us more understand the relationship of AI and labor market, the detailed analysis must be useful but cannot be explored within the limits of this paper. Therefore, we regard this issue as a limitation of our research and a gap expected to focus on in future research.

- It would be good to discuss the findings of the robustness analysis.

Response 4.4: We have tried to explain and discuss the results of the robustness tests [Lines 292-298]. The main ideas are as follows: 

On the one hand, in the previous literature, other studies have also summarized the phenomenon of wage losses for displaced workers caused by technological progress and wage premiums arose from emerging skills has already happened among many sectors, which is consistent with our findings.

On the other hand, we used “job polarization” to explain why the effects of the robustness tests on occupational employment are not significant, may the reason be that job polarization presenting U-shaped employment distribution as well known, which makes the effect of displacement risk on occupational employment failing to show significance in the linear regression.

- Minor copy-edits in a few places are needed.

Response 4.5: We have checked our manuscript throughout and edited for grammar and syntax to try our best to make sure our points clear and easy to be understood.

3.Response to Reviewer 2 Comments

Reviewer #2: The author of the article presented showed his methodology and database and obtained results that compared his hypotheses. Specifically. The author found that the regression results demonstrate that displacement risk negatively impacted occupational wage and occupational employment;suggesting that AI has a greater impact on occupational wages in higher occupational level. Therefore, I believe that the information presented has an appropriate methodological support for its publication.

Response: Thanks for the reviewer’s positive comment and encouragement. The changing in jobs and occupations caused by AI, has been a hot topic around the world nowadays. Although many countries have been concerned about the labor employment issues caused by AI technology for a long time, there are still no one-size-fits-all measures to deal with the protection of labors' rights in technological development. We still need to discuss and study more deeply and extensively for better policies.

---

## [Decision Letter · Decision Letter 1]

25 Oct 2022

Can digital skill protect against job displacement risk caused by artificial intelligence? empirical evidence from 701 detailed occupations

PONE-D-22-19557R1

Dear Dr. Tang,

We’re pleased to inform you that your manuscript has been judged scientifically suitable for publication and will be formally accepted for publication once it meets all outstanding technical requirements.

Kind regards,

Bing Xue, Ph.D.

Academic Editor

PLOS ONE

Additional Editor Comments (optional):

Reviewers' comments:

Reviewer's Responses to Questions

**Comments to the Author**

1. If the authors have adequately addressed your comments raised in a previous round of review and you feel that this manuscript is now acceptable for publication, you may indicate that here to bypass the “Comments to the Author” section, enter your conflict of interest statement in the “Confidential to Editor” section, and submit your "Accept" recommendation.

Reviewer #1: (No Response)

2. Is the manuscript technically sound, and do the data support the conclusions?

Reviewer #1: Yes

3. Has the statistical analysis been performed appropriately and rigorously? 

Reviewer #1: Yes

4. Have the authors made all data underlying the findings in their manuscript fully available?

Reviewer #1: (No Response)

5. Is the manuscript presented in an intelligible fashion and written in standard English?

Reviewer #1: Yes

6. Review Comments to the Author

Reviewer #1: Congratulations to the authors on this thorough and timely paper. My comments have largely been addressed although more could still be done to strengthen the discussion, given the importance of these recommendations. For example, in the scoping review cited under the health example (Jimenez 2020), over half the studies were from before 2005. There is literature (one example: Nazeha 2020 "A Digitally Competent Health Workforce: Scoping Review of Educational Frameworks") that cites more recent and compelling data, as well as an overview of current policies and educational efforts, that could give the authors' recommendation more weight. That said, the stated aim of this paper was "to provide empirical evidence for implementing technical and vocational education" and that aim is achieved through the analysis, so would take this recommendation as a "nice to have" rather than a prerequisite for publication.

Some other, very minor edits for grammar and syntax are suggested below:

Line 47: add "the" between "over" and "human right"

Line 99: edit "dominant role to transmit"  "dominant role in transmitting"

Line 102: add "are" between "technologies" and "utilized"

Line 106: suggest a fullstop after "usage", or syntax edit

Lines 125 - 126: suggest editing to "AI technology skills are in demand..."

Line 255: suggest change "counteraction" to "counteractive" role

Line 272: suggest removing "according to"

Line 280: suggest removing "the" before "more details"

Line 292: suggest editing to "The above results can be explained..."

Line 297: suggest editing "failing" to "fail"

Line 324/325: suggest editing "their more efficiency on information..." to "their increased efficiency with"

Line 326/327: this sentence was a bit confusinge: is it saying that 14.5% of workers with higher digital skills change jobs often, compared to 10.3% of workers with fewer digital competencies? please clarify wording.

7. PLOS authors have the option to publish the peer review history of their article (what does this mean?). If published, this will include your full peer review and any attached files.

Reviewer #1: **Yes: **Paula Ihozo Akugizibwe
